# Rate and Product Studies with 1-Adamantyl Chlorothioformate under Solvolytic Conditions

**DOI:** 10.3390/ijms22147394

**Published:** 2021-07-09

**Authors:** Kyoung Ho Park, Mi Hye Seong, Jin Burm Kyong, Dennis N. Kevill

**Affiliations:** 1Department of Chemical and Molecular Engineering, Hanyang University, Ansan-si 15588, Korea; tpdlqk@hanyang.ac.kr; 2Department of Chemistry and Biochemistry, Northern Illinois University, DeKalb, IL 60115-2862, USA; dkevill@niu.edu

**Keywords:** 1-adamantyl chlorothioformate, Grunwald–Winstein equation, solvolysis, solvolysis–decomposition, ionization pathway, carboxylium ion

## Abstract

A study was carried out on the solvolysis of 1-adamantyl chlorothioformate (1-AdSCOCl, **1**) in hydroxylic solvents. The rate constants of the solvolysis of **1** were well correlated using the Grunwald–Winstein equation in all of the 20 solvents (R = 0.985). The solvolyses of **1** were analyzed as the following two competing reactions: the solvolysis ionization pathway through the intermediate (1-AdSCO)^+^ (carboxylium ion) stabilized by the loss of chloride ions due to nucleophilic solvation and the solvolysis–decomposition pathway through the intermediate 1-Ad^+^Cl^−^ ion pairs (carbocation) with the loss of carbonyl sulfide. In addition, the rate constants (*k*_exp_) for the solvolysis of **1** were separated into *k*_1-Ad^+^Cl^−^_ and *k*_1-AdSCO^+^Cl^−^_ through a product study and applied to the Grunwald–Winstein equation to obtain the sensitivity (*m*-value) to change in solvent ionizing power. For binary hydroxylic solvents, the selectivities (*S*) for the formation of solvolysis products were very similar to those of the 1-adamantyl derivatives discussed previously. The kinetic solvent isotope effects (KSIEs), salt effects and activation parameters for the solvolyses of **1** were also determined. These observations are compared with those previously reported for the solvolyses of 1-adamantyl chloroformate (1-AdOCOCl, **2**). The reasons for change in reaction channels are discussed in terms of the gas-phase stabilities of acylium ions calculated using Gaussian 03.

## 1. Introduction

The chemistry of the acyl group and its various derivatives occupies an important place in organic and metabolic reactions [1,2]. In particular, alkyl and aryl haloformate esters [1,2,3,4,5,6] are reagents frequently used as precursors, inhibitors and protecting groups in peptide synthesis. Therefore, it is very important to study the solvolysis reactions of alkyl haloformate (alkoxy groups) and alkyl halothioformate (alkylthio groups) esters.

In previous studies, it was reported that the solvolysis reaction of 1-adamantyl chloroformate (1-AdOCOCl, **2**) [7] in hydroxylic solvents proceeds with both solvolysis–decomposition using the Grunwald–Winstein equation [8,9]. All of the reactions in **2** result in a loss of carbon dioxide to give the relatively sTable 1-adamantyl cation (1-Ad^+^Cl^−^ ion pairs). Only in 100% ethanol was a trace of 1-adamantyl ethyl carbonate (1-AdOCOOEt) observed. The picture for the solvolysis–decomposition of **2** in hydroxylic solvents is expressed on the right side of Scheme 1, with pathways involving the rate-determining unimolecular reaction with a loss of carbon dioxide. In a similar reaction, Moss et al. [10] studied product-determining intermediates on the solvolysis of 3-homoadamantyl chloroformate and 3-homoadamantyloxychlorocarbene in methanol. They demonstrated that the 3-HomoadOMe and 3-HomoadCl products are formed through an intermediate 3-Homoad^+^Cl^−^ ion pair after the loss of carbon dioxide or carbon monoxide occurs.

The solvolyses of 1-adamantyl fluoroformate (1-AdOCOF, **3**) [11], unlike the solvolytic reactions of **2** [7], were observed in two reaction pathways; one is the parallel solvolysis–decomposition pathway with a loss of carbon dioxide, as in the reaction on the right side of Scheme 1, and the other proceeds to the addition–elimination pathway including the bimolecular attack by a solvent at the acyl carbon, shown on the left of Scheme 1.

As described above, the mechanisms of the solvolyses of **2** and **3** differ depending on the solvent and leaving group. Therefore, it is interesting to study the reactivity of the tertiary bridgehead alkylthio group substituted with sulfur instead of oxygen in the alkoxy group.

Queen et al., previously reported the effects of structural changes on the rates of hydrolysis of a series of chlorothioformate esters [12,13,14]. They suggested that differences in the relative rates and reaction mechanisms of the chlorothioformate esters are due to the change in the electronic structure of the alkylthio group and the electron participation of the empty *d*-orbitals of sulfur.

Using the extended Grunwald–Winstein equation, we previously reported that the solvolysis of alkyl and phenyl chlorothioformates [15,16,17,18] in the more nucleophilic pure alcohols and aqueous solutions proceeds predominantly by an addition–elimination pathway (Scheme 1a), with an attack at the carbonyl carbon. Only in solvents of very low nucleophilicity and very high ionizing power was an ionization pathway detected (Scheme 2).

For ethyl- [16], *i*-propyl chlorothioformate (*i*-PrSCOCl, **4**) [19] and *t*-butyl chlorothioformate (*t*-butSCOCl, **5**) [17], in the majority of solvents except methanol, ethanol and 90% ethanol, the ionization pathway was dominant (Scheme 2). However, despite many studies on alkyl halothioformates, the mechanism of the tertiary bridgehead 1-adamantyl halothioformate (1-AdSCOX) has not yet been firmly established. In a previous study, the tertiary bridgehead **2** was an ionization mechanism that proceeds to the intermediate 1-Ad^+^Cl^−^ ion pair after carbon dioxide loss in all of the solvents except pure ethanol (Scheme 1b). Therefore, it is very interesting to study how the reaction of the tertiary bridgehead 1-adamantyl chlorothioformate (1-AdSCOCl, **1**) changes with a variety of solvents.

Here, we present the rate constants for reactions of **1** under solvolytic conditions, and the partitioning between the various possible solvolysis products (product selectivity, *S*). The reaction mechanism of solvolysis for **1** in a wide range of solvents is analyzed using the G–W equation. Additionally, as tools to consider the reaction mechanism for solvolysis, it deals with activation parameters (Δ*H*^≠^ and Δ*S*^≠^) and solvent isotope effects. We also consider salt effects to arrive at a reasonable mechanism. Mechanistic interpretations are compared with stabilities of the acylium ions adjacent to alkoxy and alkylthio groups, obtained by DFT calculations using Gaussian 03 [20].

## 2. Results

The rate constants of solvolysis of **1** in all of the 20 solvents at 25.0 °C are reported in Table 1, together with the solvent nucleophlicity (*N*_T_) [21,22] and solvent ionizing power (*Y*_Cl_) [23,24,25] values. Additionally, the rate constants separated into 1-Ad^+^Cl^−^ and (1-AdSCO)^+^Cl^−^ for the solvolyses of **1** at 25.0 °C are presented in Table 1. The solvents consisted of methanol (MeOH), ethanol (EtOH), 2,2,2-trifluoroethanol (TFE) and binary mixtures with water of methanol, ethanol, acetone and four binary mixtures of TFE and ethanol. Since the rates of **1** in three aqueous TFE solvents at 25 °C were too high to be measured by acid titration, the rate constants of **1** at 25 °C were obtained by extrapolation using the Arrhenius equation (Table 2).

For reactions of **1** in MeOH, EtOH, 80% EtOH and 97~70% TFE, the rate constants were also determined at three and/or four temperatures, and these data are reported in Table 2, together with data for the solvolysis of **1** in 2-propanol. These values in Table 2 were used to calculate the activation enthalpies and entropies.

The products obtained from reactions of **1** in ethanol, aqueous ethanol, aqueous acetone, TFE, aqueous TFE and TFE–ethanol mixtures have been determined using gas chromatography with a flame ionization detector. The results are reported in Table 3 and Table 4, and Scheme 3.

In Table 3 and Table 4, the small amounts of 1-AdOH found after solvolyses of **1** in 100% TFE and TFE–ethanol mixtures are probably due to the reaction of the substrate and moisture during manipulation. In the calculation of the solvent selectivity value (*S*) for binary solvent mixtures, these percentages were first subtracted from each percentage of 1-adamantanol determined before insertion into the equation, and then the selectivities were calculated (Table 3 and Table 4). For aqueous alcohol solvents and TFE-EtOH binary solvents, the selectivity values (*S*) were calculated using Equations (1)–(4), respectively.
*S*_ROH[S-D]_ = [1-AdOR] [H_2_O]/[1-AdOH] [ROH](1)
*S*_EtOH[(1-AdSCO)_^+^_Cl_^−^_]_ = [1-AdSCO_2_Et] [H_2_O]/[1-AdSH] [EtOH](2)
*S*_TFE-EtOH[S-D]_ = [1-AdOTFE] [EtOH]/[1-AdOEt] [TFE](3)
*S*_TFE-EtOH[(1-AdSCO)_^+^_Cl_^−^_]_ = [1-AdSCO_2_TFE] [EtOH]/[1-AdSCO_2_Et] [TFE](4)

In all the equations, the product concentrations are divided by the concentrations of the solvent components that produce them, and all of the concentrations are expressed as molarities. All the equations for selectivity values (*S*) are shown in the footnotes of Table 3 and Table 4, respectively.

In order to compare the salt effects of added tetraethyl ammonium chloride (Et_4_NCl), the rate constants and product percentages of the solvolysis reaction of **1** in pure ethanol at 25 °C are reported in Table 5.

## 3. Discussion

### 3.1. Kinetic Studies

The rate constants of **1** have been studied in hydroxylic solvents. In Table 1, the rates of solvolysis of **1** are similar to those obtained with solvolyses of **4** and **5** previously reported to proceed through the ionization pathway [17,19]. The tools that provide very useful information for evaluating the reaction mechanism of solvolysis reactions include the use of the simple Grunwald–Winstein (G–W) equation (Equation (5)) and the extended Grunwald–Winstein equation (Equation (6)), as follows:
log (*k*/*k*_o_) = *mY*_Cl_ + *c*(5)
log (*k*/*k*_o_) = *lN*_T_ + *mY*_Cl_ + *c*(6)

In Equations (5) and (6), *k* and *k*_o_ are the rate constants of solvolysis in a given solvent and in 80% ethanol (EtOH), respectively; *m* is the sensitivity to changes in the solvent ionizing power (*Y*_Cl_) [23,24,25], a scale of solvent ionizing power based on the rate constants of solvolysis of 1-adamantyl chloride; *l* is the sensitivity to changes in the solvent nucleophlicity (*N*_T_) [21,22], a scale of solvent nucleophilicity based on the rate constants of solvolysis of the *S*-methyldibenzothiophenium ion; and *c* is a constant (residual) term.

The correlation between the *Y*_Cl_ and *N*_T_ of the solvent was analyzed using the G–W equation for the rate constants of **1** shown in Table 1. Through these analyses, the values of *m* (a measure of bond breakage) and *l* (a measure of bond formation) were obtained. The sensitivities obtained in these correlations are reported in Table 6, together with the values previously obtained for alkyl chloroformate esters, and some sulfur-for-oxygen substituted derivatives.

An analysis using the simple G–W Equation (5) for the rate constants of **1** in all of the 20 solvents, including the values of the rate constants of the three TFE-H_2_O solvents calculated at 25 °C using the Arrhenius equation (Table 2), has a good linear correlation with values of 0.84 ± 0.03 for *m* and 0.985 for the correlation coefficient (*R*) (*F*-test value = 603) (Figure 1). For the extended G–W Equation (6), the values of 0.76 ± 0.05 for *m*, −0.15 ± 0.09 for *l* and 0.986 for the correlation coefficient (*F*-test value = 341) are obtained (Figure 2), which corresponds to a 0.087 probability that the *lN*_T_ term is not statistically significant. The *m* value for **1** is essential similar to the value (*m* = 0.73 ± 0.03) obtained for the solvolysis of **5** in all of the solvents (Table 6). This suggests that the established ionization pathway (Scheme 2) for the solvolysis of **5** also applies to **1** in this range of solvents. In Table 6, the **1** shows the negative sensitivity (−0.15 ± 0.09) to changes in solvent nucleophilicity and high sensitivity (0.76 ± 0.05) to changes in solvent ionizing power when the extended G–W equation (6) is applied. Here, the negative *l*-value (−0.15 ± 0.09) suggests even less nucleophilic participation by the solvent than in the solvolyses of 1-adamantyl chloride (*l* = 0.00, by definition) [23,24]. As described in Table 6, the correlation coefficient is almost unchanged from 0.985 to 0.986 when the simple and extended G–W equations for the rate constants of **1** in all of the 20 solvents are applied. Thus, it can be seen that the solvolyses of **1** correlate well using the simple G–W equation (5) in all of the solvent ranges studied and proceed through an ionization process similar to that of the solvolyses of **2**. That is, in the extended G–W equation for the solvolysis of **1**, the high value of *m* and the negative value of *l* are probably the result of ionization fragmentation similar to the reaction of **2** with very little assistance in nucleophilic solvation [7]. In addition, from the results obtained in the product study (Table 3 and Table 4), the experimental rate constants (*k*_exp_) for the solvolysis of **1** were separated into $k_{1-{\mathrm{Ad}}^{+}{\mathrm{Cl}}^{-}}$ and $k_{1-{\mathrm{AdSCO}}^{+}{\mathrm{Cl}}^{-}}$, and the *l* and *m* values were calculated using the G–W equation (Table 6). From these results, the *m* value of $k_{1-{\mathrm{Ad}}^{+}{\mathrm{Cl}}^{-}}$ (*m* = 0.86 ± 0.04) was higher than that of $k_{1-{\mathrm{AdSCO}}^{+}{\mathrm{Cl}}^{-}}$ (*m* = 0.64 ± 0.05). This means that in the solvolysis of **1**, the 1-Ad^+^Cl^−^ pathway is more sensitive to changes in the solvent ionizing power than the (1-AdSCO)^+^Cl^−^ pathway.

At the *m* and *l* values shown in Table 6, the primary methyl chlorothioformate [15] and ethyl chlorothioformate [16] and the secondary *i*-propyl chlorothioformate [19] previously studied proceed through dual competing reaction channels depending on various solvents, but the solvolyses of tertiary butyl chlorothioformate [17] and tertiary bridgehead 1-adamantyl chlorothioformate proceed only through the ionization pathway. Additionally, in Table 6, the *m*-values for the solvolyses of **2** and 2-adamantyl chloroformates (2-AdOCOCl, **6**, excluding data points for 100% and 90% ethanol and methanol) proceeding through the ionization pathway were also presented [26].

The activation parameter values for the solvolyses of **1** in seven solvents are tabulated in Table 2. The entropies of activation, in the range of −2.2 to −9.8 cal mol^−1^ K^−1^, are consistent with the ionization nature of the proposed rate-determining step. These values are essentially similar to those previously observed for the ionization pathway of **4** (Δ*S*^≠^ = −5.6 to −10.4 cal mol^−1^ K^−1^) and **5** (Δ*S*^≠^ = −12.2 to −14.5 cal mol^−1^ K^−1^) [17,19].

As a useful tool to investigate the reaction mechanism, it can be obtained through the solvent deuterium isotope effect caused by the isotope substitution reaction. The solvent deuterium isotopic rate ratios on the solvolysis of **1**, in methanol (*k*_MeOH_/*k*_MeOD_) and ethanol (*k*_EtOH_/*k*_EtOD_) at 25.0 °C, are shown in the footnotes of Table 1. The solvent deuterium isotopic rate ratio was obtained somewhat low with the values of *k*_EeOH_/*k*_EeOD_ = 1.05 in ethanol and *k*_MeOH_/*k*_MeOD_ = 1.35 in methanol. These are known as a characteristic of the unimolecular reaction because solvent isotope substitution does not have a significant effect on the rate. The solvent deuterium isotope effect has previously been studied for the solvolyses of haloformate esters and chlorothioformate esters. In pure methanol, the *k*_MeOH_/*k*_MeOD_ values were 1.39 for *t*-butyl chlorothioformate [17] and 1.26 for *t*-butyl fluoroformate [27], which was previously studied by the unimolecular pathway. Additionally, values in the range predicted for a bimolecular pathway were reported for *k*_MeOH/_*k*_MeOD_ of 2.17 for *n*-propyl chloroformate [28] and 2.42 for *p*-nitrobenzyl chloroformate [29], and 1.87 for **6** [26]. In general, the *k*_MeOH_/*k*_MeOD_ ratios of the ionization pathway are lower than the bimolecular pathway values. Therefore, the solvolysis reaction of **1** in all of the solvents is dominated by the ionization pathway.

### 3.2. Product Studies and Effects of Added Salt

The percentages of the observed products are reported in Table 3 and Table 4. In 100% ethanol, only 1-adamantyl ethyl thiocarbonate, 1-adamantyl ethyl ether and 1-adamantyl chloride were observed, and for solvolyses in the five aqueous-ethanol solvents, these products were accompanied by 1-adamantanethiol and 1-adamantanol.

In 100% TFE, the solvolysis products of 1-adamantyl 2,2,2-trifluoroethyl ether and 1-adamantyl chloride were observed, and in the three aqueous TFE solvents, 1-adamantanethiol and 1-adamantanol were additionally observed. There is no evidence for the formation of 1-adamantyl 2,2,2-trifluoroethyl thiocarbonate. In a previous study, for the solvolysis of **2** and **6** in the TFE–water mixtures, the percentages of 1-adamantyl chloride, formed by the loss of carbon dioxide and collapse, fall from 58.9% in 100% TFE to 19.8% in 50% TFE. However, the 1-adamantyl chloride product of solvolysis of **1** decreased slightly from 13.5 to 11.3% under the same conditions.

In 80% TFE–20% ethanol, only the two ether (1-adamantyl 2,2,2-trifluoroethyl ether and 1-adamantyl ethyl ether) and two carbonate (1-adamantyl 2,2,2-trifluoroethyl thiocarbonate and 1-adamantyl ethyl thiocarbonate) products, and 1-adamantyl chloride were observed. With a higher ethanol content, increasing amounts of 1-adamantyl ethyl thiocarbonate, formed by solvent molecular attack at the acyl carbon, were observed, and indeed, this became the dominant product in 20% TFE–80% ethanol. The 1-adamantyl chloride product obtained by the decomposition of **1** in the TFE–ethanol solvents was almost similar to the value obtained in the aqueous TFE solvents. In 100% TFE and four TFE–ethanol mixtures, a small amount of 1-adamantanol (av. 2.6%) was found (Table 3 and Table 4), and since no 1-adamantanol was found for the reaction in 100% ethanol, this would appear to be due to the presence of a small concentration of water in the TFE. This value is deducted from the reported 1-adamantanol percentages prior to the calculation of the selectivity value (*S* value). In three aqueous acetone solvents, only 1-adamantyl chloride, 1-adamantanethiol and 1-adamantanol products were observed. As described above, as the solvent ionizing power increased, the product ratios of 1-adamantyl alkyl thiocarbonate and 1-adamantanethiol decreased and the products of 1-adamantyl ethyl ether, 1-adamantyl chloride and 1-adamantanol increased. These are studied in detail by the salt effect. From the products (Table 3 and Table 4) and the values of *l* and *m* (Table 6) obtained above, it can be inferred that the solvolysis reactions of **1** in a variety of pure and binary solvents react competitively through the ionization of the R^+^ component (carbocation) and the (RSCO)^+^ component (carboxylium ion).

The overall picture of the obtained products from the reaction of the solvolysis of **1** is shown in Scheme 3. Unlike the reaction of the solvolysis of 1-adamantyl chloroformate (**2**), where only solvolysis–decomposition was observed in Scheme 1b, the solvolysis of **1**, from the *m* and *l* values of the Grunwald–Winstein equation (Table 6), the entropy of activation (Table 2) and the results of the product (Table 3 and Table 4), can be expressed as a pathway involving substitution in the carboxylium ion of acyl carbon ((1-AdSCO)^+^Cl^−^, Scheme 2) and a pathway involving the loss of carbonyl sulfide (1-Ad^+^Cl^−^, Scheme 1b).

The rate constant and product of the solvolysis of **1** in pure ethanol containing chloride ion (tetraethylammonium chloride, Et_4_NCl) are reported in Table 5. In pure ethanol, the rate gradually increases as the concentration of Et_4_NCl increases (increasing ionic strength), and the percentages of 1-adamantyl ethyl ether and 1-adamantyl chloride also increase. However, 1-adamantyl ethyl thiocarbonate decreases significantly. Therefore, when salt is added to the solvent, it is more important to proceed to the 1-Ad^+^Cl^−^ intermediate with loss of carbonyl sulfide (Scheme 1b) than to proceed to the (1-AdSCO)^+^Cl^−^ intermediate (Scheme 2). This is consistent with the proposal that a carboxylium ion attacked (solvolysis ionization pathway) by solvent at the acyl carbon is in competition with an ionic process with loss of carbonyl sulfide (solvolysis–decomposition pathway), with the ionic process preferentially favored by increases in solvent ionizing power (*Y*_Cl_).

In Table 3 and Table 4, the nature of the solvolysis product of **1** has been studied for aqueous ethanol and for mixtures of TFE with water or ethanol. In aqueous ethanol, the selectivity (*S*) values represented by the two competition reactions slightly increase from 0.45 to 0.58 for the reaction proceeding to 1-Ad^+^Cl^−^ and decrease from 0.81 to 0.52 for (1-AdSCO)^+^Cl^−^ as the water content increases. In aqueous TFE, the selectivity (*S*) values gradually increase from 0.78 to 2.12 as the TFE content decreases. These values are very similar to the values observed from reactions of **2** (*S*_EtOH-H2O_ = 0.60~0.88 and *S*_TFE-H2O_ = 1.03) [7], **6** (*S*_TFE-H2O_ = 0.62~0.99) [26] and benzyl chloroformate (*S*_TFE-H2O_ = 1.05~1.20) [29] believed to be the unimolecular reaction (1-Ad^+^Cl^−^ ion pair) with the loss of carbon dioxide (Scheme 1b).

As the ethanol content in the TFE–ethanol mixture increases, the selectivity for TFE relative to ethanol indicates a smaller preference shown by bridgehead adamantyl derivatives for attack by TFE molecules than ethanol molecules (Table 4). These selectivity values (*S*) decrease from 2.58 to 1.62 for the reaction proceeding to 1-Ad^+^Cl^−^ and from 1.97 to 1.41 for (1-AdSCO)^+^Cl^−^. As shown in Scheme 1b and Scheme 3, and Table 4, all of the products of the solvolysis of **1** in TFE–ethanol mixtures can be represented by the following two competition reactions: an ionization pathway involving the loss of a chloride ion to give a carboxylium ion with appreciable stabilization by nucleophilic solvation, and a solvolysis–decomposition pathway with the loss of carbonyl sulfide. From these results, in solvents of high ionizing power and relatively low nucleophlicity, it is preferentially favored that the solvolysis reaction of **1** proceeds to the 1-Ad^+^Cl^−^ pathway than proceeding through (1-AdSCO)^+^Cl^−^.

In the solvolysis of **1** in aqueous alcohol, unlike the reactions of the solvolysis of **2**, the products of 1-AdSH and 1-AdSCOOR were additionally obtained through intermediate (1-AdSCO)^+^Cl^−^. These results appear to be due to the fact that in the resonant hybrid (stabilization by the positive mesomer effect) formed with the initial acyl ions in the transition state, sulfur can transfer positive charges better than oxygen [14,19,30]. That is, because the sulfur atom supports a positive charge better than oxygen, and oxygen exerts a strong affinity for electrons in the R-O bond. All the structures for **1** and **2** were fully optimized using the Gaussian 03 and density functional theory (DFT) calculations at the 6–31G(d) level by B3LYP, and then the charge, distance and energy between the atoms were calculated. Sulfur has a smaller electronegativity and a larger radius than oxygen. Therefore, the induction effect on carbonyl carbon is less; therefore, the carbonyl carbon around sulfur has a smaller positive charge (1-AdSC*=O, electric charge (EC) = 0.143, 1-AdOC*=O, EC = 0.478) (Appendix A, Appendix A). From the above experiments, it was confirmed that the rate-determining step of solvolysis of **1** proceeds as a unimolecular ionization reaction. It is possible to predict the reaction pathway that forms a resonance-stabilized carboxylium ion intermediate, a resonance structure in which electrons of sulfur of alkyl chlorothioformate are easily transferred to the carbonyl carbon then formed a π bond (Scheme 2). On the other hand, in **2**, which is a tertiary bridgehead structure, since the electronegativity of the oxygen atom of alkoxy groups is large, the resonance structure of the alkoxy groups (1-AdO*-C=O)^+^ is somewhat more unstable than the alkylthio groups (1-AdS*-C=O)^+^ in transferring electrons to the carbonyl carbon. Therefore, in the solvolysis of **2**, after carbon dioxide loss becomes decomposition, 1-Ad^+^Cl^−^ ion pairs are formed, and the solvolysis reaction proceeds to obtain a more stable reaction. It is very similar to the proposed process for N_2_O loss for the solvolysis of substituted benzyl azoxyarenesulfonates reaction by Maskill [31], and for carbon dioxide or carbon monoxide loss for the solvolysis of 3-homoadamantyl chloroformate and 3-homoadamantyloxychlorocarbene reaction by Moss [10].

## 4. Materials and Methods

The synthesis of 1-adamantyl chlorothioformate (1-AdSCOCl, **1**) was prepared similarly to the synthesis of 1-adamantyl chloroformate (1-AdOCOCl, **2**) as previously described [7]. The 1-adamantanethiol (1-AdSH, Sigma-Aldrich, St. Louis, MO, USA) was used as received. All of the substrates were recrystallized from ligroin before using. Solvents were purified as described previously [7]. The rates for the solvolytic reaction of **1** were followed using the potentiometric titrations method, and the kinetic experiments and the products were carried out as described previously [29]. The rate constants were obtained by averaging all of the values from, at least, duplicate runs. The products from the reactions of **1** under solvolytic conditions were analyzed after 10 half-lives by gas chromatography (GC-9A, Shimadzu, Kyoto, Japan) using a 2.1-m glass column containing 10% Carbowax 20M on ChromosorbWAW80/100, as previously described (Appendix A) [29]. Simple and multiple regression analyses were carried out using the OriginPro 2016 from the OriginLab Corporation.

## 5. Conclusions

The solvolysis of **1** gives a good linear correlation (R = 0.985 and *F* value = 603) using the simple (one-term) G–W Equation (5) in all of the solvent ranges studied, and the *m* value (0.84 ± 0.03) of **1** is moderately similar to the values obtained for the solvolyses of **3** and **5** demonstrated by the ionization mechanism (Table 6). The reactions of the solvolysis–decomposition ionization pathway, 1-Ad^+^Cl^−^, and the solvolysis ionization pathway, (1-AdSCO)^+^Cl^−^, were separated from the experimental rate constant (*k*_exp_) of the solvolysis of **1**. From these results, it was found that the reaction of the intermediate 1-Ad^+^Cl^−^ (*m* = 0.86 ± 0.04) was more sensitive (*m*-value) to changes in the solvent ionizing power (*Y*_Cl_) than the reaction of (1-AdSCO)^+^Cl^−^ (*m* = 0.64 ± 0.05) (Table 6). Additionally, the values for the entropies of activation in the range of −2.2 to −9.8 cal mol^−1^ K^−1^ and the solvent deuterium isotope rate ratio, *k*_EtOH_/*k*_EtOD_ = 1.05 and *k*_MeOH_/*k*_MeOD_ = 1.35 are consistent with the natures of ionization reaction.

In addition to 1-adamantyl alkyl thiocarbonate (1-AdSCOOR) and 1-adamantanethiol (1-AdSH), the solvolysis products of **1** have the following three additional types of products formed with the loss of carbonyl sulfide: 1-adamantyl chloride (1-AdCl, decomposition product), 1-adamantanol (1-AdOH) and 1-adamantyl alkyl ether (1-AdOR). Compared to the product of **2**, these additional types of formation, 1-AdSCOOR and 1-AdSH, are favored in solvents of high nucleophilicity and/or low ionizing power. The products formed after the carbonyl sulfide loss are very similar in character to those formed from **2**. The selectivity values (*S*) of **1** are a moderately similar to the values observed from reactions of **2**, **6** and benzyl chloroformate, believed to be the ionization reaction with the loss of carbon dioxide (intermediate 1-Ad^+^Cl^−^, Scheme 1b). Additionally, in solvents of high nucleophilicity and low ionizing power, **1** forms a more stable carboxylium ion ((1-AdSCO)^+^, Scheme 2) than **2** because it can transfer a positive charge better than oxygen in a resonance hybrid in which sulfur is formed as an initial acyl ion.

In the present study, the solvolysis reaction of **1** in all of the solvents confirmed that there are the following two major ionization pathways: a parallel solvolysis–decomposition ionization pathway (Scheme 1b) involving the loss of carbonyl sulfide (1-Ad^+^Cl^−^), such as the solvolysis of **2**, and a solvolysis ionization pathway (Scheme 2) in which a nucleophilic solvent reacts with a resonance-stabilized carboxylium ion, (1-AdSCO)^+^.

## Data Availability

“MDPI Research Data Policies” at https://www.mdpi.com/ethics.

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
