# Peer review of "Rate and Product Studies with 1-Adamantyl Chlorothioformate under Solvolytic Conditions"

_ijms, 2021, doi:10.3390/ijms22147394_

Round 1
Reviewer 1 Report
In this manuscript the authors deal with mechanism of solvoysis of 1-Adamantyl Chlorothioformate. This research is in some form continuation of previous researches of mechanisms of solvolysis of different haloformates, as 1-adamantyl chloroformate, and halotioformates.
Authors proposed mechanism of solvolysis reaction of 1-Adamantyl Chlorothioformate based on Grunwald-Winstein equation, product analysis, solvent isotope effect, salt effects and activation parameters. They explained their conclusions based on the obtained results so I recommend this paper for publication but there are a few suggestions and things that need to be corrected:
Page 1, line 15 – “decompostion“ should be “decomposition“
Page 1, line 41 – “carbon dioxide“ instead of “carbon oxide“
Page 2, line 49 – „through“ instead of „though“
Page 2, line 73 – t-buSCOCl should be t-butSCOCl
Page 2, line 76-78 the part of sentece („...the tertiary bridgehead 2 was an ionization mechanism that proceeds...“) should be preformulated because of clarity.
Page 3, line 85 – considered instead od consider
Page 6, line 180 – below the Table 6., „s“ is the same as „n“ (in line 178)
Page 7, line 185 – in the description of the Figure 1. is written that the abscissa is a Y-value, but it is not well marked in the picture. It should be corrected because of clarity.
Page 7, line 187 - in the description of the Figure 2. is written that the abscissa is a -0.15NT+0.76Y(Cl), but it is not well marked in the picture. It should be corrected because of clarity.
Page 8, line 232 – Instead of Table 2, probably it should be Table 1 (there is no solvent deuterium isotopic rate ratios in the footnotes of Table 2.)
Page 9, line 290 – It should be Table 5 instead of Table 4.
Page 9, line 296 – “attacked” instead of “attack”
Page 10, lines 300-301 – In Table 4 there are only results connected to TFE-ethanol, not for “…aqueous ethanol and for mixtures of TFE with water or ethanol.”
Page 10, line 321 – it should be corrected 1-ASCOOR in 1-AdSCOOR
In Table 1 and 4 there are in a few places written T (when authors write about TFE-ethanol mixtures), because of clarity it should be corrected to TFE.
Author Response
To Reviewer 1
Thank you so much for your brief and kind review of our paper. I corrected all your review contents as you showed me. Line 185 and 187 you mentioned, I think reviewer version had some problems. The manuscript I downloaded in order to revise, it should be shown the abscissa in Figure 1 and 2. And you asked to correct T to TFE in Table 1 and 4, I checked Table 1 has the brief contents about it in footnote i, and I added the same footnote in Table 4.
Again I appreciate for your nice reviewing.
Sincerely,
Kyoung-Ho Park
Reviewer 2 Report
The authors investigated kinetically the solvolysis mechanism of 1-adamantyl chlorotioformate in the series of aqueous and pure solvents. Based on GW m values, products and their ratios, solvent KIE, activation parameters, selectivity etc. the authors demonstrated that the solvolysis of title compounds proceeds via two parallel pathways, decomposition-ionization and ionization pathway, respectively.
This is a nice piece of work, the experiments are carefully designed and the conclusions are straightforward. I think that this work is appropriate to be published in IJMS.
I have some minor suggestions which will enable easier reading of the ms.
Throughout the text I had problem with realizing which substrate is discussed. The number is given only in the text beside the name of the compound, and the structure is not presented with formula. It makes reading difficult. For example, on the beginning of Discussion (pg 6) the authors refer to structures 4 and 5. Since I did not remember which structures have these numbers I had to go back into Introduction and search the text (I found the data at the bottom of pg 2). I suggest the authors to present the substrates and some important reference structure on a scheme to enable easy reading.
In Figures 1 and 2 the values of the abscissa the variable are not shown. What is "R"? Also, since the GW m value is discussed, it would be illustrative to show the correlation line.
I the text (pg 8, line 232) it is stated that the solvent KIEs are given in the footnotes of Table 2. However, these data are missing.
Since I do not have a data for KIEs, I could not understand the following sentence (pg 8, lines 232-234): "The solvent deuterium isotopic rate ratio was obtained somewhat lower with the values of kEeOH/kEeOD=1.05 in ethanol and kMeOH/kMeOD=1.35 in methanol". Lower of what? What is compared?
Finally, I find that important information is missing from the Abstract - the established reaction mechanisms. I strongly suggest the authors to add one sentence in the Abstract in which the mechanisms for solvolysis of the title compound established in this work are presented (similarly as in the last sentence in Conclusions).
Author Response
To Reviewer 2
Thank you so much for your brief and kind review of our paper. I tried to correct your review results.
Additionally, I would like to explain about your review results.
First, the numbers for indicating the substrates, is usual expression in our papers. I think you also know that so many names and many times of substrates are shown on our paper. So we are using the represent number for the substrate in order to make the confident paper for readers. I will discuss it with editor, I will try it
Second, KIEs are in Table 1, so I corrected that. And, if the values of KIEs are close to unity, we can say that the values of KIEs is low. So I switch the word “lower” to “low”.
Third, in abstract, I mentioned the mechanism of solvolysis-decomposition which is one of the unimolecular pathway. I think this is very important reaction mechanism of 1-adamantyl chlorothioformate in our this time study.
Again I appreciate for your nice reviewing.
Sincerely,
Kyoung-Ho Park
Reviewer 3 Report
In this manuscript, the authors have described the rate and products of the tertiary bridgehead 1-adamantylthiochloroformate with respect to changes in the nucleophilicity, and Ionizing power of different solvents.
Here are my comments:
1) Please correct the structure of adamentyl group in Scheme 1. Please make sure the bond angles are correct.
2) Please correct the typo in Line 66, Page 2 (Alchohol).
3) Introduce NT and Ycl before using abbreviations.
Given the novelty and interest in scientific community, I recommend the editors to accept this manuscript for publication.
Author Response
To Reviewer 3
Thank you so much for your brief and kind review of our paper. I corrected your review results.
Again I appreciate for your nice reviewing.
Sincerely,
Kyoung-Ho Park